# Predictors of Outcome after (Chemo)Radiotherapy for Node-Positive Oropharyngeal Cancer: The Role of Functional MRI

**DOI:** 10.3390/cancers14102477

**Published:** 2022-05-18

**Authors:** Pasqualina D’Urso, Alessia Farneti, Laura Marucci, Simona Marzi, Francesca Piludu, Antonello Vidiri, Giuseppe Sanguineti

**Affiliations:** 1Department of Radiotherapy, IRCCS Regina Elena National Cancer Institute, 00144 Rome, Italy; alessia.farneti@ifo.it (A.F.); laura.marucci@ifo.it (L.M.); giuseppe.sanguineti@ifo.it (G.S.); 2Medical Physics Laboratory, IRCCS Regina Elena National Cancer Institute, 00144 Rome, Italy; simona.marzi@ifo.it; 3Radiology and Diagnostic Imaging Department, IRCCS Regina Elena National Cancer Institute, 00144 Rome, Italy; francesca.piludu@ifo.it (F.P.); antonello.vidiri@ifo.it (A.V.)

**Keywords:** oropharyngeal cancer, chemoradiotherapy, DWI, DCE-MRI

## Abstract

**Simple Summary:**

We have conducted a prospective study on patients with locally advanced oropharyngeal cancer who are candidates for concomitant radio-chemotherapy; we considered their anamnestic findings, tumor characteristics and evaluated the role of innovative radiological features, particularly the magnetic resonance imaging (MRI) biomarkers. Our aim was to identify those elements correlated with worse tumor control. Diffusion-weighted (DWI) imaging and dynamic contrast–enhanced (DCE) can help identify hypoxic regions in head and neck cancer which are known to be more resistant to the effects of radiation. A better understanding of these factors may help us improve our knowledge on tumor behavior and thus provide a more tailored treatment in patients that respond poorly.

**Abstract:**

The prognosis of a subset of patients with locally advanced oropharyngeal cancer (LA-OPC) is still poor despite improvements in patient selection and treatment. Identifying specific patient- and tumor-related factors can help to select those patients who need intensified treatment. We aimed to assess the role of historical risk factors and novel magnetic resonance imaging (MRI) biomarkers in predicting outcomes in these patients. Patients diagnosed with LA-OPC were studied with diffusion-weighted imaging (DWI) and dynamic-contrast enhanced MRI at baseline and at the 10th radiotherapy (RT) fraction. Clinical information was collected as well. The endpoint of the study was the development of disease progression, locally or distantly. Of the 97 patients enrolled, 68 were eligible for analysis. Disease progression was recorded in 21 patients (11 had loco-regional progression, 10 developed distant metastases). We found a correlation between N diameter and disease control (*p* = 0.02); features such as p16 status and extranodal extension only showed a trend towards statistical significance. Among perfusion MRI features, higher median values of K_ep_ both in primary tumor (T, *p* = 0.016) and lymph node (N, *p* = 0.003) and lower median values of v_e_ (*p* = 0.018 in T, *p* = 0.004 in N) correlated with better disease control. K_ep_ P90 and N diameter were identified by MRMR algorithm as the best predictors of outcome. In conclusion, the association of non-invasive MRI biomarkers and patients and tumor characteristics may help in predicting disease behavior and patient outcomes in order to ensure a more customized treatment.

## 1. Introduction

The initial approach in patients with locally advanced head and neck squamous cell carcinoma (HNSCC) is a combination of radiotherapy and concomitant chemotherapy (RT-CT) in order to avoid surgery and to preserve organ functionality. However, the outcome of these patients is still poor, with 40–50% experiencing disease recurrence [1,2]. Patient- and tumor-related predictors of response to RT-CT would be helpful for a more efficient selection of candidates to non-surgical approach and to avoid unnecessary toxicity in non-responders. In the last decades, some major changes have occurred in the epidemiology and management of these patients, especially those with oropharyngeal cancer (OPC), and new prognostic factors have been identified. Currently, HPV-positive OPC is considered a distinct entity that is usually associated with a younger age of onset, distinct clinical features, limited tobacco exposure and a more favorable oncological outcome than HPV-negative OPC [3,4]. Histologically, it is characterized by basaloid, lymphoepithelial and poorly differentiated features [5], and radiologically by cystic-appearing lymph nodes on both magnetic resonance imaging (MRI) and computed tomography (CT).

However, despite the better prognosis, a significant proportion of these patients continues to experience distant disease progression [6,7]. Historical risk factors such as significant smoking history and extranodal extension (ENE) in lymph nodes maintain a significant predictive role of an increased risk of both locoregional (LRF) and distant failure [8,9,10,11].

Response to treatment is also influenced by biological (rather than anatomical) features such as intrinsic tumor radiosensitivity and hypoxia [12,13]. Advances in MR imaging techniques currently allow the estimation of tissue cell density and the localization of hypoxic regions within HNSCC using novel functional biomarkers: diffusion-weighted (DWI) imaging quantifies the water molecule mobility in tissues which is strictly related to cell architecture, while dynamic contrast–enhanced (DCE) MRI allows for the derivation of semi-quantitative hemodynamic maps by the estimation of the passage of blood through vessels [14].

The aim of this study is to prospectively assess by a machine learning (ML) approach the role of both traditional and emerging/novel factors in predicting outcomes in patients with OPC who underwent (chemo)radiotherapy.

## 2. Materials and Methods

### 2.1. Patient Population and Treatment Characteristics

We evaluated 97 consecutive patients treated within a non-randomized, prospective, single-institution trial funded by the Italian Association for Cancer Research (AIRC, project No.17028), and specific informed consent was obtained before enrollment.

Inclusion criteria to be met were: (a) age older than 18 years; (b) histologically-confirmed squamous cell carcinoma of the oropharynx; (c) locally advanced tumor stages III and IV according to 8th edition of the AJCC Cancer Staging Manual [15]; (d) definitive treatment with concomitant RT-CT. Exclusion criteria considered were: any contraindication for MR examination due to previous allergic reaction to intravenous contrast material administration or renal disease; patients previously treated with surgery, chemotherapy or radiotherapy to the head and the neck. Disease staging required MRI examination and CT of the neck and chest or PET-CT for distant staging. Baseline characteristics, including smoking pack-years and alcohol consumption, were collected as well. Patients received intensity modulated radiation therapy (IMRT) with a simultaneous integrated boost technique in 35 fractions over seven weeks. Prescribed doses were 70 Gy to macroscopic primary (T) and lymph node disease (N), and 63 Gy and 58.1 Gy to areas at high risk and at low risk of subclinical disease, respectively [16]. Concomitant chemotherapy consisted of cisplatin, three times weekly (100 mg/m^2^ for three cycles every 21 days) or weekly (40 mg/m^2^ for 6 cycles) [17].

### 2.2. MRI Protocols

MRIs were done on a 1.5 T scanner (Optima™ MR450w, GE Healthcare, Milwaukee, WI, USA) with a head and neck phased-array coil. All patients underwent three serial studies: at baseline, at the 10th fraction of RT and eight weeks after RT. MRI follow-up examinations were performed every six months for the first two years, and once per year afterwards. The pretreatment MRI protocol included coronal and axial T2-weighted images (field of view, 26–28 cm; acquisition matrix, 288 × 256; slice thickness, 4 mm), intra-voxel incoherent motion diffusion-weighted imaging (IVIM-DWI), and dynamic-contrast enhanced MRI (DCE-MRI) sequences. IVIM-DWI was performed using nine b values (b = 0, 25, 50, 75, 100, 150, 300, 500, and 800 s/mm^2^, field of view 26 × 28 cm; acquisition matrix, 128 × 128; slice thickness, 4 mm; scanning time, 6 min 13 s). DCE-MRI was performed using a 3D fast-spoiled gradient echo sequence (flip angle, 30°; field of view, 28 cm; acquisition matrix, 128 × 128; slice thickness, 4 mm; spacing between slices, 2 mm; temporal resolution, 5 s; scanning time, 5 min). Three pre-bolus phases were acquired, followed by contrast-arrival phases after the intravenous administration of gadolinium-based contrast agent, at a rate of 3 mL/s (60 total dynamic phases). To reduce the use of contrast medium, only IVIM-DWI was performed during treatment.

### 2.3. DCE-MRI and DWI Analysis

3D Slicer Software (version 4.6.2) was used for the lesion visualization and segmentation [18]. Commercial software (GenIQ General, GE Advanced Workstation, Palo Alto, CA, USA) was used to derive the quantitative maps from the DCE-MRI on the basis of a two-compartment pharmacokinetic model and automatic population-based selection of the arterial input function [19]. Three perfusion parameters were calculated at the single voxel level: K^trans^, defined as the transfer constant between plasma and the extravascular extracellular space (EES), and K_ep_, defined as the transfer constant between EES and plasma and v_e_, which represents the fractional EES volume. The baseline contours of the lesions, both T and the largest N, were performed on T2-weigheted images and, after rigid propagation, were transferred on the corresponding perfusion maps to perform quantitative analyses. The medians, percentiles (P) P10, P25, P75, and P90, skewness, kurtosis, energy, and entropy values were calculated from the voxel-based distribution of perfusion parameters within the entire lesion. The bin size used for each patient to extract the data was 0.2 min^−1^, 0.4 min^−1^ and 0.025 for K^trans^, K_ep_ and v_e_, respectively. The lesions at baseline and at the 10th fraction were also outlined on DWI at b = 800 s/mm^2^ to extract the diffusion coefficients from the signal intensity curve at increasing b values, by means of home-made scripts developed in MATLAB (Release 2020b, MathWorks Inc., Natick, MA, USA). To reduce the instability of the diffusion-weighted signal within single voxels and increase the robustness of the quantifications, the median of the signal from the entire lesion at each b value was extracted and used for the fitting process. The conventional ADC was derived from data at b values of 0, 500, and 800 s/mm^2^, while the perfusion-free diffusion coefficient Dt was derived from data at b values of 300, 500, and 800 s/mm^2^, by a mono-exponential function [20]. The Levenberg–Marquardt algorithm was used to perform the fits. The perfusion fraction f was derived from an asymptotic estimate through an extrapolation of the signal intensity S0extr to b = 0 s/mm^2^ from the above calculation of Dt: f = [(S0meas − S0extr)/S0meas] × 100. S0meas indicates the measured signal intensity at b value of 0 s/mm^2^ [21].

### 2.4. Statistics

The endpoint of the study is the development of disease progression at the primary site, neck or distantly, alone or in combination at a follow up time of two years. Patients experiencing second primary tumors or death due to intercurrent disease who precluded a two year minimum follow up time were excluded.

Since an integrated statistical approach is suggested to improve data interpretability and prediction accuracy [22], data analyses were approached with both conventional statistical methods (to infer the relationships between variables) and ML algorithms (to at best design the prediction model). Regarding the former, univariate analyses on disease progression were performed considering various patient-, tumor- and treatment-related variables (p16 status, smoking habit, alcohol abuse, T subsite, T size, lymph node diameter, matted lymph nodes, cystic lymph nodes, ENE). Groups were compared with the chi-squared test or the Mann–Whitney rank test when appropriate. Statistical significance was claimed for *p* values < 0.05.

### 2.5. Machine Learning Analysis

Before the model building, among the above mentioned ones, the parameters with the best classification performance were selected using the minimum redundancy maximum relevance (MRMR) algorithm. The selection of the most informative predictors is a separate and mandatory step of the ML analysis pipeline to avoid the model overfitting and reduce to the minimum the subset of variables to make the best predictions, without loss of information [23]. The MRMR algorithm ranks both categorical and continuous parameters for classification and it finds the optimal set that was mutually and maximally dissimilar, based on the mutual information of variables [24]. The most common ML algorithms were quickly trained on our dataset to compare their performances and find the most appropriate [23]. Details are available in the Appendix A. The model classification ability was evaluated in terms of accuracy, sensitivity, specificity, positive predictive value (PPV) and negative predictive value (NPV) after a stratified five-fold cross-validation to prevent overfitting and improve the possibility of generalizing the models. The prediction accuracies between different models were compared using the mid-*p*-value McNemar test. The ML analysis was carried out using the MATLAB environment.

## 3. Results

### 3.1. Patient Characteristics

Out of the 97 enrolled patients, 68 (70.1%) were considered eligible. Eight patients without lymph node involvement were excluded. Twenty-one patients were not considered evaluable for disease response due to death due to intercurrent disease (N = 17) or to disease (N = 4) within 24 months from treatment end. There were 53 male patients (77.9%) and the median age was 61 years (IQR: 55–69 years). Selected patient and tumor characteristics are summarized in Table 1. Median follow-up was 33.2 months (IQR: 26.3–46.9 months). The majority of tumors (N = 53, 77.9%) were p16 positive. The distribution of primary subsites within the oropharynx was significantly different in p16 positive and negative patients (Table 1, *p* = 0.012). Similarly, smoking status and alcohol consumption were also distributed differently according to p16 status (*p* = 0.041 and *p* < 0.001 for smoking and drinking, respectively).

P16 negative disease presented with a slightly smaller lymph node disease (median 2.5 cm vs. 1.6 in p16 negative patients, *p* = 0.06). Five patients refused chemotherapy and were treated with RT alone. At a median follow up of 9.7 months (IQR: 8.5–15.5 moths), disease progression was observed in 21 patients (30.8%). Interestingly, all the failures had been observed within two years from treatment end, and none afterwards. Regarding the pattern of disease progression, 10 patients (47.6%) developed distant metastases (lung metastasis in eight patients, bone in two patients); 11 patients (52.4%) had loco-regional disease progression (eight patients in lymph nodes, one in primary disease, three patients in both). Predictors for disease control following univariate analysis are shown in Table 2. Among all covariates, a significant correlation with disease control was found only for N diameter (*p* = 0.02). P16 status and ENE had a marginal impact on disease control (*p* = 0.134 and 0.073, respectively).

### 3.2. MRI Analysis and Prediction Models Driven by Machine Learning

The summary statistics of DCE-MRI and DWI parameters are reported in Table 3, Table 4 and Table 5.

Among radiological features, higher median values of K_ep_ and lower median values of v_e_ at both the primary tumor (*p* = 0.016 and *p* = 0.018 for K_ep_ and v_e_, respectively) and the lymph nodes (*p* = 0.003 and *p* = 0.004 for K_ep_ and v_e_, respectively) were associated with better disease control. Diffusion parameters did not correlate with clinical outcome, with the exception of ΔADC (%), which showed a trend towards significance (*p* = 0.073). The MRMR algorithm identified K_ep_ P90 and the diameter of N as best predictors among all the categorical and continuous parameters included in the analyses (Appendix A). The performance of different families of ML algorithms is reported in Appendix A, showing the superiority of the Decision Tree classification learner on our dataset with respect to the other tested algorithms. The model accuracy of the Decision Tree classifier was 83.8% (95%CI: 72.9–91.6%), with Sensitivity, Specificity, PPV and NPV values of 61.9% (95%CI: 38.4–81.9%), 93.6% (95%CI: 82.5–98.7%), 81.3% (95%CI: 58.0–93.2%) and 84.6% (95%CI: 76.0–90.5%), respectively. To better infer the role of DCE-MRI, we explored the predictive performances obtained with different combinations of the most relevant parameters for comparison with the best model (Appendix A). As illustrated in Appendix A, the best model is the one which incorporates nodal K_ep_ P90 along with baseline N diameter. All other models lacking DCE-parameters were statistically inferior to the one including K_ep_ P90 in terms of accuracy and/or specificity and/or sensitivity (Appendix A).

Two representative clinical cases are shown in Figure 1.

## 4. Discussion

In the present study, we found that baseline (perfusion) MRI features have an independent predictive role of disease outcome at two years at conventional statistics. Moreover, with a ML approach, the model incorporating selected DCE-MRI parameters along with clinical ones was the best predictor of outcome in terms of accuracy, specificity or sensitivity. Therefore, perfusion MRI features add to the clinical ones in predicting oncologic outcome after (chemo)radiotherapy for OPC. Recently, there has been an increasing interest in baseline or pre-treatment quantitative imaging features [25,26] with the purpose of customizing and further refining treatment strategies [27]. Similarly, to other recent investigations [28,29,30], we found several DCE-based parameters at both the primary tumor and the nodes to be correlated with outcome. In particular, higher median values of K_ep_ and lower median values of v_e_ were found to be associated with better tumor control rates. This is consistent with the fact that hypoxic tumors usually need more aggressive treatments [31] and are at higher risk of treatment failure [28]. In our dataset, the Decision Tree classification learner provided the best accuracies with respect to the other common ML algorithms. The histogram analysis of perfusion maps and the calculation of percentiles seemed to improve the capability to identify relationships between DCE-MRI parameters and outcome. Indeed, the model incorporating the parameter K_ep_ P90, along with the nodal diameter, was the best predictor for disease progression among those tested (Appendix A), providing a good accuracy, a high specificity and a fair sensitivity. These results are consistent with the current literature investigating prognostic functional imaging in patients with squamous cell carcinoma of the head and neck. In the review of Bos et al. [30], higher K_ep_ values were associated to a higher probability of both overall survival and loco-regional control. Moreover, K_ep_ was also identified as an independent predictor of disease free survival. Based on the pharmacokinetic model [19], the parameter K_ep_ is positively associated to K^trans^ but compared to K^trans^, it should be less influenced by blood volume/flow and more influenced by vessel permeability and the characteristics of the extravascular and extracellular compartments [32]. In our study, patients with disease control showed only a trend towards higher K^trans^ (median) values while v_e_ values were significantly lower in both primary tumors and lymph nodes. Being derived from the ratio K^trans^/ v_e_, K_ep_ was able to amplify these differences, showing a superiority compared to the other DCE-derived parameters. Concerning DWI features, we assumed that the perfusion-free diffusion coefficient Dt could be more appropriate to quantify the thermal diffusivity of water molecules in tissue and, consequently, to better indirectly evaluate the tissue cellular density and its early radiation-induced modifications compared to ADC. However, neither ADC nor Dt were significantly different between patients with disease control and progression, except for the nodal percentage variation of ADC during treatment. This parameter showed a trend towards significance, suggesting that a smaller increase in ADC was indicative of a higher risk of progression in accordance with previous investigations [33,34]. With regard to patient and tumor characteristics, it is a matter of fact that survival in patients with HNSCC is influenced by the presence of nodal metastases [35,36,37]. In HPV-related disease, there is a proven association between the infection and neck lymph-node disease [38]. Our finding is in line with the well-known clinical evidence; advanced N-disease staging is a well-established risk factor for disease progression [9]. Patients with high burden neck disease, particularly the presence of a matted node, have a higher risk of developing distant metastasis [39,40]. Lymph node size is also tightly correlated with regional recurrence [41]. Among morphological characteristics, we found a significant association between the nodal diameter and disease control, with a higher risk of progression for patients with larger pathologic lymph nodes. Concerning HPV status and ENE, we found a trend towards worse disease control in patients with a higher proportion of HPV-negative OPC and with ENE, though the difference was not statistically significant. Both HPV negative status and ENE are well-established factors of poor prognosis [42,43]. In the retrospective analysis of Ang et al. [3], patients with HPV-related disease showed a better three-year overall survival compared to HPV-negative patients, though factors such as heavy smoking history and advanced T and N stage had an independent detrimental effect on outcome. Despite a large number of HPV-positive patients, the vast majority of patients were smokers, thus limiting the benefits of HPV status. Similarly, even if the presence of ENE has been acknowledged as a poor predictor of outcome after radiotherapy for head and neck squamous cell carcinomas [10] and along with positive surgical margin, it represents an indication for chemo-radiotherapy in the post-operative setting [44]; its role in patients with HPV-positive disease is less clear. In the systematic review by Benchetrit et al. [45], radiological and pathological ENE in HPV-positive OPC leads to higher distant recurrence and worse overall survival but had a limited impact on locoregional recurrence. The present study has some limitations. In order to limit the impact of confounding factors on outcome (i.e., death from intercurrent causes), we selected only patients with a 2-yr minimum follow up. This strengthened the potential relationships between the various covariates and the oncologic outcome, though it reduced the sample size and thus the power of the study. Moreover, we focused only on a particular subset of head and neck cancer, oropharynx, mostly of which were actually virus-related. Therefore, generalization of the present findings to other head and neck subsites may not be appropriate. The accuracy of v_e_ and K_ep_ estimations may have been affected by the scan time of the DCE-MRI sequence. Indeed, a scan duration ≥10 min is recommended to allow the contrast agent to reach the equilibrium after leaking into the EES and leaking back to the vascular space [46]. However, such a long scan time is difficult to be used in clinical practice, particularly in head and neck cancer patients, due to their limited tolerance; moreover, the variability in DCE-MRI acquisition protocols and image post-processing in the literature has prevented, up to now, a harmonization between results from different institutions, limiting the replicability of our data. Furthermore, we did not include analyses on ADC/Dt maps to reduce the instability of the diffusion-weighted signal within single voxels; in future investigations we could explore the potential of a histogram-based approach to diffusion maps for enhancing the performance of the classification model.

## 5. Conclusions

Perfusional features at baseline MRI such as K_ep_ and v_e_ are predictive of tumor response at two years in patients affected by locally advanced oropharynx cancer and, when incorporated in a model along with clinical factors, help to build the predictive model with the highest performance in predicting clinical outcome. Therefore, DCE-MRI findings can potentially provide additional and unique information to offer more personalized treatment. Further studies on a larger population are needed to confirm our observations.

## Figures and Tables

**Figure 1 cancers-14-02477-f001:**
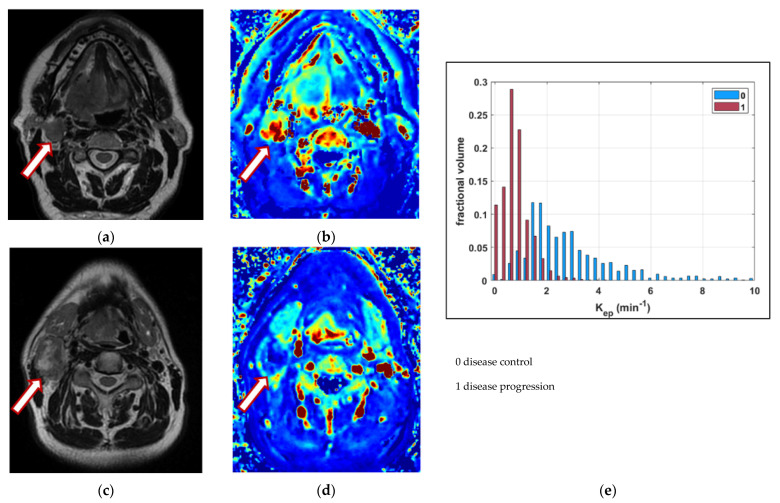
A case of a 69-year-old male patient affected by a carcinoma of the base of the tongue, with disease control. T2-weighted image (**a**) and the corresponding K_ep_ map (**b**) show a solid and well-perfused pathologic lymph node in the right cervical region (indicated by the arrow); at the bottom, a case a 60-year-old male patient affected by a carcinoma of the base of the tongue, without disease control. T2-weighted image (**c**) and the corresponding K_ep_ map (**d**) show a large and necrotic lymph node. The comparison of the K_ep_ distribution in the entire N volume for the patients with and without disease control is depicted (**e**).

**Table 1 cancers-14-02477-t001:** Patients’ characteristics.

Characteristic	p16-Pos (#/%)	p16-Neg (#/%)	Overall (#/%)
cN1	33 (62.3%)	2 (13.3%)	35 (51.5%)
cN2	17 (32%)	5 (33.3%)	22 (32.3%)
cN2a	-	1	
cN2b	-	4	
cN2c	-	0	
cN3	3 (5.7%)	8 (53.4%)	11 (16.2%)
cN3a	-	0	
cN3b	-	8	
Unilateral N	38 (71.6%)	11 (73.3%)	49 (72%)
Bilateral N	15 (28.4%)	4 (26.7%)	19 (28%)
ENE present	35 (66%)	6 (40%)	41 (60.3%)
ENE not present	18 (34%)	9 (60%)	27 (39.7%)
Cystic N on MR yes	23 (43.4%)	5 (33.3%)	28 (41.2%)
Cystic N on MR no	30 (56.6%)	10 (66.4%)	40 (58.8%)
Cystic N on CT yes	21 (39.6%)	4 (26.7%)	25 (37.8%)
Cystic N on CT no	32 (60.4%)	11 (73.3%)	43 (63.2%)
Matted nodes present	12 (22.6%)	2 (13.3%)	14 (20.6%)
Matted nodes not present	41 (77.4%)	13 (86.7%)	54 (79.4%)
Subsite			
Tonsil	30 (56.6%)	3 (20%)	33 (48.5%)
Base of tongue	23 (43.4%)	12 (80%)	35 (51.5%)
Smoking status			
none	10 (18.9%)	0	10 (14.7%)
0–5 pack/year	18 (33.9%)	3 (20%)	21 (30.9%)
6–24 pack/year	7 (13.3%)	1 (6.7%)	8 (11.8%)
>24 pack/year	18 (33.9%)	11 (73.3%)	29 (42.6%)
Alcohol			
None	40 (75.5%)	5 (33.3%)	45 (66.2%)
Social	12 (22.6%)	2 (13.3%)	14 (20.6%)
Alcoholic	1 (1.9%)	8 (53.4%)	9 (13.2%)
Overall	53 (77.9%)	15 (22.1%)	68 (100%)

#: number.

**Table 2 cancers-14-02477-t002:** Univariate analysis.

Variable	Disease Progression		
	No	Yes	*p* value
P16			
Yes	39	14	0.134
No	8	7	
T subsite			
Tonsil	22	138	0.253
Base of tongue	25		
Smoking habit			
No	9	1	0.122
0–5 pack/year	15	6	
6–24 pack/year	5	3	
>24 pack/year	18	11	
Alcohol			
None	34	11	0.253
Social	7	7	
alcoholic	6	3	
T size * (cm)	3.4 (2.6–4.2) *	3.4 (1.5–3.4) *	0.942
N diameter * (cm)	1.8 (1.6–2.5) *	2.8 (2.1–4.2) *	0.022
ENE present			
Yes	25	16	0.073
No	22	5	
Matted lymph nodes			
Yes	8	6	0.276
No	39	15	
Cystic lymph nodes on MR			
Yes	22	6	0.158
No	25	15	
Cystic lymph nodes on CT			
Yes	19	6	0.349
No	28	15	

* Median (95% confidence interval).

**Table 3 cancers-14-02477-t003:** Summary statistics of DCE-MRI parameters in the primary tumor (T) for patients with and without disease control.

		Disease Control		Disease Progression		
DCE-MRI Parameter		Median	IQR	Median	IQR	*p* Value
**K^trans^**	*Median*	0.80	0.39	0.63	0.47	0.158
	*IQR*	0.50	0.43	0.44	0.39	0.204
	*P10*	0.42	0.15	0.36	0.33	0.496
	*P25*	0.61	0.24	0.46	0.44	0.246
	*P75*	1.06	0.68	0.92	0.71	0.204
	*P90*	1.45	1.24	1.25	1.48	0.260
	*Skewness*	1.77	1.44	1.88	1.10	0.900
	*Kurtosis*	6.95	9.09	10.82	11.37	0.271
	*Mean*	0.92	0.60	0.81	0.66	0.294
	*Std*	0.62	0.52	0.48	0.46	0.223
	*Energy*	0.15	0.10	0.18	0.13	0.160
	*Entropy*	3.16	1.02	2.84	1.04	0.193
**K_ep_**	*Median*	2.24	0.96	1.80	0.64	**0.016**
	*IQR*	1.60	1.48	1.20	1.04	0.151
	*P10*	1.04	0.40	0.84	0.72	**0.035**
	*P25*	1.52	0.56	1.24	0.80	**0.008**
	*P75*	3.16	1.76	2.36	1.28	**0.034**
	*P90*	4.80	3.12	3.28	3.04	0.137
	*Skewness*	5.02	4.97	8.05	10.95	0.120
	*Kurtosis*	56.18	107.39	112.25	404.49	0.120
	*Mean*	2.91	1.44	1.99	1.35	**0.033**
	*Std*	2.47	2.68	1.93	1.50	0.193
	*Energy*	0.21	0.20	0.36	0.22	0.063
**v_e_**	*Entropy*	2.62	1.16	1.99	1.29	0.094
	*Median*	0.38	0.12	0.43	0.12	**0.018**
	*IQR*	0.14	0.06	0.16	0.10	0.257
	*P10*	0.23	0.10	0.28	0.15	0.364
	*P25*	0.30	0.10	0.35	0.09	0.051
	*P75*	0.45	0.15	0.51	0.17	**0.018**
	*P90*	0.52	0.16	0.59	0.33	**0.014**
	*Skewness*	0.18	0.96	0.28	0.96	0.573
	*Kurtosis*	4.57	2.03	3.83	2.98	0.434
	*Mean*	0.38	0.13	0.43	0.13	**0.016**
	*Std*	0.12	0.05	0.13	0.09	0.507
	*Energy*	0.07	0.03	0.06	0.03	0.405
	*Entropy*	4.18	0.57	4.36	0.79	0.415

Statistically significant *p*-values are bold. *p* values refer to Mann-Whitney test. Abbreviations: K^trans^ (min^−1^), transfer constant between plasma and EES (extravascular extracellular space); K_ep_ (min^−1^), transfer constant between EES and plasma; v_e_, fractional volume of EES.

**Table 4 cancers-14-02477-t004:** Summary statistics of DCE-MRI parameters in the lymph node (N) for patients with and without disease control.

		Disease Control		Disease Progression		
DCE-MRI Parameter		Median	IQR	Median	IQR	*p* Value
**K^trans^**	*Median*	0.55	0.38	0.45	0.37	0.189
	*IQR*	0.42	0.31	0.30	0.35	0.069
	*P10*	0.24	0.30	0.21	0.19	0.582
	*P25*	0.38	0.32	0.32	0.24	0.536
	*P75*	0.80	0.50	0.66	0.60	0.069
	*P90*	1.14	0.81	0.83	0.76	**0.036**
	*Skewness*	2.16	1.81	2.02	1.98	0.393
	*Kurtosis*	10.00	14.24	10.87	13.27	0.951
	*Mean*	0.66	0.44	0.51	0.39	0.065
	*Std*	0.44	0.25	0.37	0.24	**0.036**
	*Energy*	0.19	0.08	0.25	0.23	0.109
	*Entropy*	2.80	0.78	2.36	1.28	0.077
**K_ep_**	*Median*	1.84	1.12	1.28	0.96	**0.003**
	*IQR*	1.28	0.96	0.80	0.60	**0.000**
	*P10*	0.80	1.16	0.64	0.44	0.446
	*P25*	1.20	0.80	0.96	0.84	**0.044**
	*P75*	2.48	1.56	1.76	1.20	**0.000**
	*P90*	3.52	2.66	2.24	1.64	**0.000**
	*Skewness*	3.97	4.82	4.12	2.45	0.604
	*Kurtosis*	28.81	79.47	42.11	46.95	0.795
	*Mean*	2.22	1.30	1.39	1.26	**0.000**
	*Std*	1.86	2.05	0.81	1.07	**0.001**
	*Energy*	0.28	0.14	0.42	0.19	**0.002**
**v_e_**	*Entropy*	2.16	0.84	1.53	0.96	**0.001**
	*Median*	0.27	0.20	0.40	0.24	**0.004**
	*IQR*	0.16	0.05	0.16	0.13	0.272
	*P10*	0.13	0.21	0.22	0.13	0.140
	*P25*	0.20	0.20	0.31	0.15	**0.007**
	*P75*	0.36	0.22	0.49	0.31	**0.004**
	*P90*	0.45	0.22	0.62	0.38	**0.008**
	*Skewness*	0.58	0.78	0.08	0.93	**0.018**
	*Kurtosis*	4.26	2.82	3.84	1.59	0.328
	*Mean*	0.28	0.20	0.42	0.23	**0.003**
	*Std*	0.12	0.05	0.14	0.08	**0.036**
	*Energy*	0.07	0.03	0.06	0.05	0.228
	*Entropy*	4.12	0.46	4.35	0.73	0.069

Statistically significant *p*-values are bold. *p* values refer to Mann-Whitney test. Abbreviations: K^trans^ (min^−1^), transfer constant between plasma and EES (extravascular extracellular space); K_ep_ (min^−1^), transfer constant between EES and plasma; v_e_, fractional volume of EES.

**Table 5 cancers-14-02477-t005:** Summary statistics of the diffusion parameters and their variations at the 10th fraction in primary tumor (T) and lymph node (N), for patients with and without disease control.

Diffusion Parameter	Disease Control		Disease Progression		
T	Median	IQR	Median	IQR	*p* Value
ADC	1.34	0.47	1.31	0.50	0.480
D_t_	1.00	0.31	1.00	0.35	0.882
f (%)	15.60	7.53	14.80	10.48	0.462
ADC_10fr_	1.74	0.34	1.75	0.42	0.968
D_t,10fr_	1.38	0.32	1.39	0.33	0.377
f_10fr_ (%)	18.14	9.67	16.12	6.27	0.345
ΔADC (%)	33.20	46.40	33.20	27.65	0.503
ΔD_t_ (%)	32.30	48.68	28.00	24.40	0.558
Δf (%)	13.65	74.15	31.00	84.75	0.382
**N**	Median	IQR	Median	IQR	*p* value
ADC	1.09	0.37	1.19	0.39	0.640
D_t_	0.96	0.38	0.93	0.41	0.700
f (%)	9.75	6.00	8.80	5.23	0.895
ADC_10fr_	1.50	0.43	1.43	0.44	0.447
D_t,10fr_	1.24	0.49	1.24	0.28	0.938
f_10fr_ (%)	12.47	6.72	9.87	3.80	0.228
ΔADC (%)	32.80	31.00	20.95	32.05	0.073
ΔD_t_ (%)	33.40	36.48	23.75	34.80	0.268
Δf (%)	25.55	80.20	2.60	75.00	0.211

*p* values refer to Mann-Whitney test. Abbreviations: ADC (×10 − 3 mm^2^/s), apparent diffusion coefficient; Dt(×10 − 3 mm^2^/s), tissue diffusion coefficient; f (%). ΔADC, ADC variation (%) relative to the pretreatment value (analogously for the other parameters), Δ: delta.

## Data Availability

Data Availability Statement: The original contributions presented in the study are included in the article/Appendix A. Further inquiries can be directed to the corresponding author.

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
