# Peer review of "Predictors of Outcome after (Chemo)Radiotherapy for Node-Positive Oropharyngeal Cancer: The Role of Functional MRI"

_cancers, 2022, doi:10.3390/cancers14102477_

Round 1
Reviewer 1 Report
1) Authors prospectively assessed the role of historical risk factors and novel magnetic resonance imaging biomarkers (IVIM-DWI and DCE-PWI) in predicting outcomes of 68 patients with advanced oropharyngeal cancer treated with (chemo)radiotherapy. Results of the study are clearly described, and they may be useful in clinical practice. Authors found that higher median Kep and lower median Ve values on baseline (pretreatment) DCE-PWI at both the primary tumor and the lymph nodes were associated with a better disease control. For IVIM-DWI, ADC variations (%) relative to pretreatment value showed a trend towards clinical outcome. The only thing I would like to ask the authors is if they can better briefly clarify in the discussion the role of the Ktrans parameter in their study. Authors assesed that Kep is positively associated to Ktrans and (rightly, ndr) that Kep is less influenced than Ktrans by blood volume/flow. May author simply clarify this sentence based on the results obtained in the study, as they did for Kep ad Ve?
2) Please fix errors in Table 1; some contents (maybe "0"), % and a round bracket are missing for cN stages.
Author Response
Thanks for the comment.
1) We have modified the discussion as it follows:
"Based on the pharmacokinetic model [19], the parameter Kep is positively associated to Ktrans but compared to Ktrans, it should be less influenced by blood volume/flow and more influenced by vessel permeability and the characteristics of the extravascular and extracellular compartments [43]. In our study, patients with disease control showed only a trend towards higher Ktrans (median) values while ve values were significantly lower in both primary tumors and lymph nodes. Being derived from the ratio ktrans/ ve, Kep was able to amplify these differences, showing a superiority compared to the other DCE-derived parameters.
2) Errors reported have been fixed.
Reviewer 2 Report
Overall Comments
This is an interesting and well done study. The authors describe in detail each variable. The methodology is well explained and performed, all aspects in the methdology were clearly explained in the results section, which is adequately included in the tables with each included variable. I consider in this interesting study but the authors need make supervival studies related with RT in which the relationship with MRI protocols, DCE-MRI and DWI Analysis are analyzed, showing Kaplan-Meier analysis related to HPV and with other risk factors (Tabaquism and Alcoholism). Moreover, they evaluate progressive diseases, stable disease, period of locoregional recurrence, period before metastasis and treatment performed.Author Response
Thanks for the comment. However, since all the included patients have a 2-yr minimum follow up, and thus they should be at low risk of failure after this time point, we decided to approach the analysis ignoring the time factor and with a binary endpoint consisting of presence/absence of disease within 2 years from treatment end. Therefore, we minimized the confounding impact of censored events (such as death due to intercurrent causes within 2 yrs) and we were able to investigate the relationships between the most relevant clinical characteristics and specific MRI features. Within the present statistical approach, survival curves, though interesting, cannot be estimated.